# What Constrains Adaptation After Pretraining? Representation Dynamics Under Inherited Data Manifolds

## Abstract

Large language models are often adapted after pretraining under fixed objectives and training pipelines, yet their behavior across data settings can be difficult to anticipate. We study whether post pretraining adaptation is primarily constrained by optimization and supervision, or by geometric structure inherited from pretraining. Using controlled data selection, we treat training samples as structured populations in representation space and compare adaptation under matched architectures and training procedures while varying only geometric support. Across the controlled models and continued pretraining settings we study, adaptation mainly redistributes density within already occupied regions, with limited cross-region migration.

## 1 Introduction

Applied systems must choose between reuse across settings and specialization for specific needs Caruana (1997). The difficulty becomes clear when shared representations are asked to support different and often conflicting requirements. Modern language models offer a particularly clear case. A single pretrained backbone is often reused across tasks and domains, with measurable behavior and representation statistics varying substantially across settings Raffel et al. (2020).

Modern language models derive their capabilities from vast training corpora, yet research attention has focused disproportionately on models rather than on data itself Bommasani et al. (2021). Over the past few years, the dominant training pipeline has largely stabilized. Pretraining on large collections of public text corpora Peters et al. (2018) is followed by post pretraining adaptation, typically continued pretraining under the same next token objective, and in some pipelines further alignment stages Ouyang et al. (2022). We use continued pretraining as a controlled probe of post pretraining adaptation, and we do not model alignment. These stages are now well understood and widely adopted across model families.

Model behavior is often interpreted through objectives and optimization, while the organization of the training data remains largely unexamined. Training data are then processed as collections of samples, without regard to their arrangement in representation space. Correspondingly, a large body of work has framed progress in terms of model scaling Kaplan et al. (2020); Hoffmann et al. (2022), objective design van den Oord et al. (2019), and optimization algorithms Kingma & Ba (2015); Loshchilov & Hutter (2019). While the geometric organization of representations has received comparatively less explicit attention Rogers et al. (2020).

Related work has shown that learned representations have internal structure, and that this structure constrains how models behave Bronstein et al. (2017). Earlier work on distributed representations shows that behavior depends on how internal states are organized, not on isolated symbolic rules Rumelhart & McClelland (1987). From this perspective, the effects of adaptation are best examined through the geometry of representation space, rather than solely through coarse behavioral evaluations Elhage et al. (2021).

In this work, we treat training data as a structured population embedded in representation space. In our setting, adaptation redistributes density within inherited regions, with local neighborhoods becoming more uniform under highly overlapping support.

## 2 Related Work

### Pretraining: Foundation of Language Models

Pretraining establishes the representation space on which later adaptation operates. Through large scale exposure to diverse text corpora, transformer models learn representations that capture broad linguistic structure Vaswani et al. (2017); Tennenholtz et al. (2024). This process organizes data into a structured representation space, beyond improvements in surface level task performance.

Prior work in representation learning shows that learned representations reflect both data structure and training objectives, rather than forming a uniform embedding space Bengio et al. (2013). Studies of neural population dynamics further suggest that high dimensional systems evolve along lower dimensional manifolds, which constrain subsequent learning dynamics Sussillo & Barak (2013); Cunningham & Yu (2014). Consistent with this view, adaptation after pretraining often reuses existing structure, with global organization persisting while metric properties such as angular separation can sharpen Raghu et al. (2017).

### Beyond Pretraining: Adaptation and Its Constraints

Adaptation after pretraining updates model parameters while operating on representations that are already established. Distillation is another common route, typically matching a teacher through output distributions Sun et al. (2019), intermediate representations Jiao et al. (2020); Wang et al. (2020), or attention patterns Tian et al. (2020).

Across these settings, several analyses suggest that post pretraining updates tend to reuse and reshape existing structure rather than creating entirely new representational organization Mirzadeh et al. (2020); Hao et al. (2022). We use continued pretraining updates (full-parameter and LoRA) as a controlled probe and vary only the geometric support of the training data.

### Data Geometry in Representation Space

Training pipelines rarely make explicit how training data are situated in representation space. Data are typically processed as unordered samples, with geometry left implicit Bommasani et al. (2021). Even when data are treated as a first class concern, they are typically handled through filtering, sampling, or efficiency driven choices, rather than through explicit control of representation geometry Wang et al. (2023); Albalak et al. (2024). Empirical studies of pretraining data Longpre et al. (2024) composition similarly emphasize domain coverage, data age, and quality, while leaving the geometric organization of data in representation space implicit.

A substantial body of prior work has shown that learned representations reside on structured, non Euclidean manifolds rather than forming uniform clouds of points Bronstein et al. (2017). Classical manifold learning frameworks highlight that data geometry constrains the scope of downstream transformations Roweis & Saul (2000); Tenenbaum et al. (2000); Belkin & Niyogi (2003). Similar constraints have been observed in recent analyses of embedding organization and mechanistic structure in large language models Nanda et al. (2023).

Adaptation after pretraining operates under an inherited geometric structure. Training data drawn from different regions of the manifold induce distinct representation dynamics, even when training settings are held fixed.

## 3 Experimental Design

To make the comparisons clean, we keep the following fixed: *(i)* architecture and pretrained initialization, *(ii)* training budget and input processing, *(iii)* update operator (full parameter updates or LoRA updates) and its hyperparameters, and *(iv)* the evaluation protocol and representation extraction procedure (Appendix A). Unless otherwise noted, *adaptation* refers to continued pretraining under the standard next token causal language modeling objective. We treat full parameter updates and LoRA updates as two update operators

for the same adaptation setting. In this paper, we use the term fine tuning as shorthand for continued pretraining, rather than instruction supervised fine tuning.

## 3.1 Experimental Setup

**Models and Adaptation.** We use the pretrained decoder only language models `meta-llama/Llama-3.2-1B` and `meta-llama/Llama-3.2-3B` Grattafiori et al. (2024). We evaluate two update operators under matched optimization settings: full parameter updates and LoRA based updates, both under continued pretraining. Let $f_\theta$ denote a pretrained decoder only language model with parameters $\theta$. Unless otherwise noted, all adaptation updates start from the same pretrained checkpoint of $f_\theta$.

**Dataset Construction.** Training data come from the Simplified-NQ Train dataset (Wikipedia-derived text). After normalization, we keep sequences between 150 and 250 tokens. For multilingual experiments, we build matched-budget datasets of total size $N_{\text{train}} = 9000$: English only (9000), English+Chinese (4500 each), English+Chinese+Hindi (3000 each), and English+Chinese+Hindi+Arabic (2250 each). All sampling uses a fixed random seed of 42.

**Training Configuration.** We train with the standard next-token causal language modeling objective using packed sequences of length 300. Unless noted otherwise, models are trained for 1 epoch with learning rate $2 \times 10^{-5}$, per-device batch size 4, warmup ratio 0.03, weight decay 0, and random seed 42.

## 3.2 Geometry Conditioned Data Construction and Metrics

**Geometry Evaluation.** For neighborhood statistics, we mean-pool final-layer embeddings over non-padding tokens. Each condition is evaluated on $N_{\text{eval}} = 1000$ held-out sequences (seed 42). We compute Euclidean $k$-nearest-neighbor statistics with $k = 10$, including mean, standard deviation, coefficient of variation, and the 90th percentile distance. We also report an effective-dimension proxy using the participation ratio $d_{\text{eff}} := (\sum_j \lambda_j)^2 / \sum_j \lambda_j^2$, where $\{\lambda_j\}$ are the nonnegative eigenvalues of the covariance matrix computed from the mean-pooled final-layer embeddings $\{z_i\}_{i=1}^{N_{\text{eval}}}$ in the decoder only model space.

**Dataset Geometry and Conditioned Sampling** Rather than treating the dataset as an unordered collection of samples, we treat candidate samples as a structured population embedded in a fixed reference embedding space.

**Reference embedding space.** Dataset geometry is defined using a frozen sentence embedding encoder, `BAAI/bge-m3`. Let $g_{\text{ref}}$ denote this frozen reference encoder. Let $\mathcal{X}_{\text{pool}} = \{x_i\}_{i=1}^{N_{\text{pool}}}$ denote the candidate pool, where $N_{\text{pool}}$ is the number of candidate sequences. For each $x_i$, define its reference embedding as $u_i = g_{\text{ref}}(x_i)$ using mean pooling over tokens. Let $\mathcal{Z}_{\text{ref}} = \{u_i\}_{i=1}^{N_{\text{pool}}}$ denote the resulting reference embeddings. The fixed encoder yields a consistent space for density estimation and sampling, independent of the representations of $f_\theta$ that evolve during adaptation. Because $g_{\text{ref}}$ is used only to define a fixed sampling rule, its tokenization and pooling choices do not affect any reported geometric measurements, which are computed exclusively in the model space of $f_\theta$.

**kNN radius proxy and quantile sampling.** Local neighborhood radius is approximated using the mean $k$-nearest neighbor distance:

$$\rho_k^{\text{ref}}(u_i) = \frac{1}{k} \sum_{u_j \in \mathcal{N}_k^{\text{ref}}(u_i)} \|u_i - u_j\|_2. \tag{1}$$

Unless otherwise noted, we use $k = 10$ for all reference-space density estimates and quantile partitions. Paired datasets are constructed under identical budgets using fixed quantile thresholds. Let $Q_\alpha^{\text{ref}}$ denote the $\alpha$-quantile of the empirical distribution of $\{\rho_k^{\text{ref}}(u_i)\}_{i=1}^{N_{\text{pool}}}$ in the reference space. We fix the quantile levels

as $\alpha_{\mathrm{c}} = 0.2$ and $\alpha_{\mathrm{p}} = 0.8$ throughout all experiments. We then set $q_{\mathrm{low}} \coloneqq Q_{\alpha_{\mathrm{c}}}^{\mathrm{ref}}$ and $q_{\mathrm{high}} \coloneqq Q_{\alpha_{\mathrm{p}}}^{\mathrm{ref}}$, and define

$$\begin{aligned} \mathcal{D}_{\mathrm{center}} &= \{x_i : \rho_k^{\mathrm{ref}}(u_i) \leq q_{\mathrm{low}}\}, \\ \mathcal{D}_{\mathrm{periph}} &= \{x_i : \rho_k^{\mathrm{ref}}(u_i) \geq q_{\mathrm{high}}\}. \end{aligned} \tag{2}$$

For quantile ties, we rank by $\rho_k^{\mathrm{ref}}(u_i)$ and take equal-sized subsets from both tails. Samples are drawn from $\mathcal{D}_{\mathrm{center}}$ and $\mathcal{D}_{\mathrm{periph}}$. The two sets are matched in size and preprocessing (formatting, tokenization, packing, and objective). Within each paired comparison (center vs. periphery), the partition rule in the reference space is the only change.

**Rationale for a separate reference space.** We define dataset geometry and all sampling rules in the frozen reference space of `BAAI/bge-m3`. All reported geometry and adaptation dynamics are evaluated only in the decoder only model space of $f_\theta$. This separation decouples data selection from representations that change during adaptation.

**Sanity check (reference–model agreement).** On 10,000 sequences, the reference-space density ranking correlates with the pretrained model-space density ranking (Spearman $\rho_{\mathrm{S}} = 0.714$, $p < 10^{-10}$; $k = 10$). See Appendix B.

**Role of visualization** UMAP visualizations are included solely to provide qualitative context. All conclusions are based on quantitative statistics that are invariant to scale, which are reported in later sections.

### 3.3 Language and Domain Structure in Representation Space

We focus on two sources of annotation, **language** and **domain**. As shown in Fig. 1, language labels correspond to well separated regions in representation space, whereas coarse domain labels occupy substantially overlapping regions. This contrast motivates the hypothesis examined in this section. We investigate whether adaptation dynamics differ when representations occupy separable versus entangled regions of the inherited manifold.

### 3.4 Evaluation Metrics

**Objective fit.** We report token-level negative log likelihood, denoted by NLL, on a held-out split processed with the same pipeline as training.

**Geometric evaluation.** Representation geometry is assessed using statistics computed from final-layer embeddings. Global measures include $\mathbb{E}[\|z\|_2]$, $\mathbb{E}[d_2]$, $\mathbb{E}[d_{\cos}]$, $\mathbb{E}[d_2^{\mathrm{dir}}]$ (direction only Euclidean distance). All expectations $\mathbb{E}[\cdot]$ are empirical averages over the evaluated sequences, with pairwise terms estimated by subsampling.

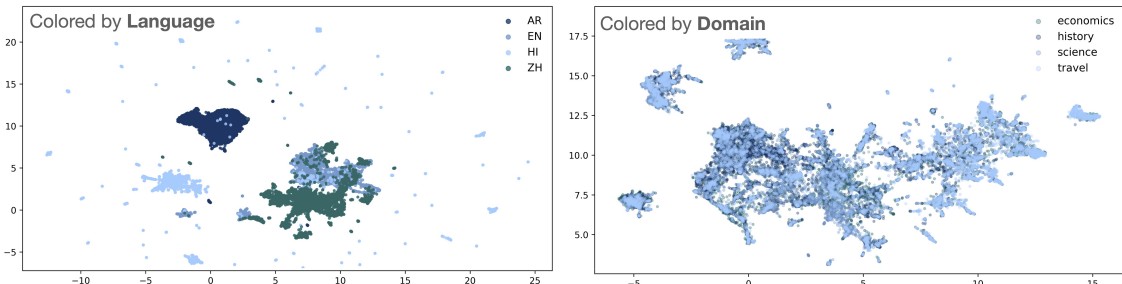

Figure 1: **Dataset geometry under different annotation axes.** UMAP is computed from last-layer embeddings of the pretrained `meta-llama/Llama-3.2-1B` model. **(a)** Embeddings colored by language, showing well separated regions. **(b)** The same embeddings colored by coarse domain labels, which exhibit substantial overlap.

**Pairwise subsampling protocol.** Let $N_{\text{eval}}$ denote the number of evaluated sequences, and let $\mathcal{Z} = \{z_i\}_{i=1}^{N_{\text{eval}}}$ be the corresponding embeddings. To avoid $O(N_{\text{eval}}^2)$ computation, we estimate global pairwise statistics by sampling $M = 200{,}000$ index pairs $(i_m, j_m)$ with $i_m, j_m \in \{1, \ldots, N_{\text{eval}}\}$ and enforcing $i_m \neq j_m$. We use a fixed random seed of 42 for pair sampling and reuse the same sampled index pairs across checkpoints, models, and training operators. We report

$$\texttt{global\_intra} := \frac{1}{M} \sum_{m=1}^{M} \|z_{i_m} - z_{j_m}\|_2, \qquad \texttt{std} := \text{Std}\Big(\{\|z_{i_m} - z_{j_m}\|_2\}_{m=1}^{M}\Big).$$

For each $z_i$, let $\mathcal{N}_k(z_i) \subset \mathcal{Z} \setminus \{z_i\}$ be its $k$ nearest neighbors under Euclidean distance. We define the mean neighbor distance

$$\rho_k(z_i) := \frac{1}{k} \sum_{z_j \in \mathcal{N}_k(z_i)} \|z_i - z_j\|_2.$$

We report $\texttt{knn\_mean} = \mathbb{E}[\rho_k]$, $\texttt{knn\_std} = \text{Std}(\rho_k)$, $\texttt{knn\_cv} = \texttt{knn\_std}/\texttt{knn\_mean}$, and $\texttt{knn\_q90} = Q_{0.9}(\rho_k)$, where $Q_p(\rho_k)$ denotes the $p$-quantile of the empirical distribution of $\{\rho_k(z_i)\}_{i=1}^{N_{\text{eval}}}$.

**Representation Extraction for Geometry Analysis** Representations are taken from the final layer for each input sequence $x$ and each training checkpoint. Unless otherwise noted, we represent a sequence by mean-pooling last-layer hidden states using the tokenizer attention mask to exclude padding positions, yielding $z(x) \in \mathbb{R}^d$. Additional statistics are computed on $L_2$ normalized embeddings to separate scale sensitive effects from directional structure.

**Baselines and Scope** The goal of this study is to isolate the effect of geometric support in representation space. We therefore vary only geometry-conditioned sampling while keeping other aspects of data selection fixed. Methods such as random sampling, quality filtering, curriculum learning, or similarity-based heuristics change multiple factors simultaneously and would confound this analysis.

## 4 Experiments and Evaluation

### 4.1 Early Geometric Effects of Post Pretraining Adaptation

We first examine how full parameter continued pretraining alters the geometry of representation space in a multilingual setting. Fig. 2 presents joint fit Uniform Manifold Approximation and Projection (UMAP) visualizations McInnes et al. (2020) of final layer embeddings extracted from a 1B parameter model before adaptation (baseline) and after four epochs of language conditioned full parameter continued pretraining (Epoch 4).

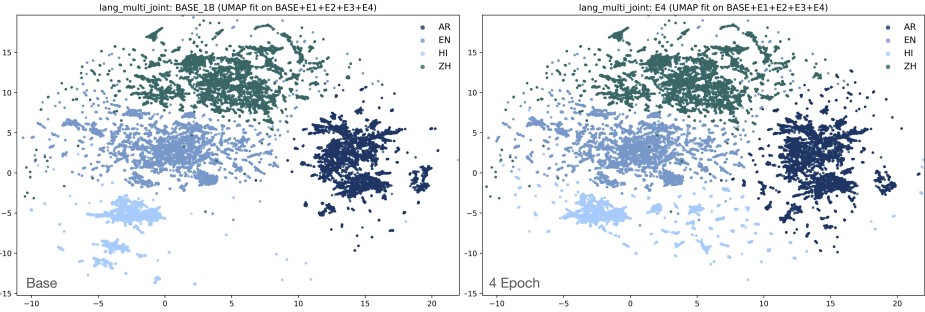

Figure 2: **Early geometric effects of language conditioned continued pretraining.** Joint fit UMAP projections of final layer embeddings from a 1B model at baseline and after four epochs of full parameter updates. Language clusters remain separated while density redistributes within each cluster, with no cross language collapse.

| Epoch | Mean norm | Norm std | Mean L2 dist | Mean cosine dist | Mean dir. dist |
|---|---|---|---|---|---|
| Base | 61.826 | 4.942 | 71.167 | 0.716 | 1.146 |
| Epoch 1 | 59.356 | 6.103 | 69.819 | 0.740 | 1.169 |
| Epoch 2 | 55.628 | 5.388 | 66.116 | 0.750 | 1.182 |
| Epoch 3 | 53.550 | 5.139 | 64.028 | 0.757 | 1.190 |
| Epoch 4 | 52.929 | 5.095 | 63.378 | 0.759 | 1.192 |

Table 1: **Changes in global representation geometry across continued pretraining epochs (language conditioned, 1B model).** Mean embedding norm, global Euclidean distance (`global_intra`), cosine distance, and direction only Euclidean distance computed from last layer embeddings.

UMAP projections reveal stable relative positions of language clusters after full parameter continued pretraining, alongside redistribution of density within clusters. We assess global distance statistics across training epochs to clarify whether the observed behavior reflects representational collapse or a contraction in scale (Table 1). Pairwise expectations are estimated using $M = 200{,}000$ randomly sampled pairs with a fixed random seed shared across epochs.

Across training epochs, embeddings contract in scale. The mean norm drops from 61.826 at baseline to 52.929 after four epochs of continued pretraining, accompanied by a parallel decrease in average Euclidean pairwise distance from 71.167 to 63.378. This contraction does not reflect a loss of directional structure. Instead, cosine based global distances increase from 0.716 to 0.759, and direction only Euclidean distances computed on normalized embeddings rise from 1.146 to 1.192. Together, these trends indicate that angular separation between representations is preserved and slightly increases under this setting.

Taken together, the joint fit UMAP visualizations and global distance statistics indicate scale contraction with preserved, and slightly increased, angular separation. Under language conditioning, continued pretraining primarily redistributes density within existing language regions while preserving their separation, with directional structure becoming slightly sharper. This motivates an axis dependent analysis of how adaptation alters geometry under different annotation axes.

### 4.2 Stability under Extended Continued Pretraining

Adaptation is continued for up to twenty epochs under the same language-conditioned setting, allowing changes observed after a small number of epochs to be examined over a longer training horizon. Fig. 3 presents joint fit UMAP projections of final layer embeddings from the same 1B model, covering checkpoints from the baseline through epochs 19. A shared projection and fixed axis limits are used across checkpoints.

Separability is quantified using a centroid based cosine margin (Appendix C) for language identification. Top one language retrieval remains perfect across all checkpoints, indicating that language clusters do not collapse. The mean margin decreases monotonically from 0.705 at baseline to 0.630 at epochs 19 and stabilizes after approximately epochs 13. This stabilization behavior is visualized in Fig. 4. This trend reflects a gradual reduction in inter language separation while preserving overall cluster separability. Detailed statistics for individual languages are reported in Appendix C.

Across training, language clusters maintain their relative positions. Density shifts occur within individual language regions, the relative organization is preserved, while density is redistributed within regions.

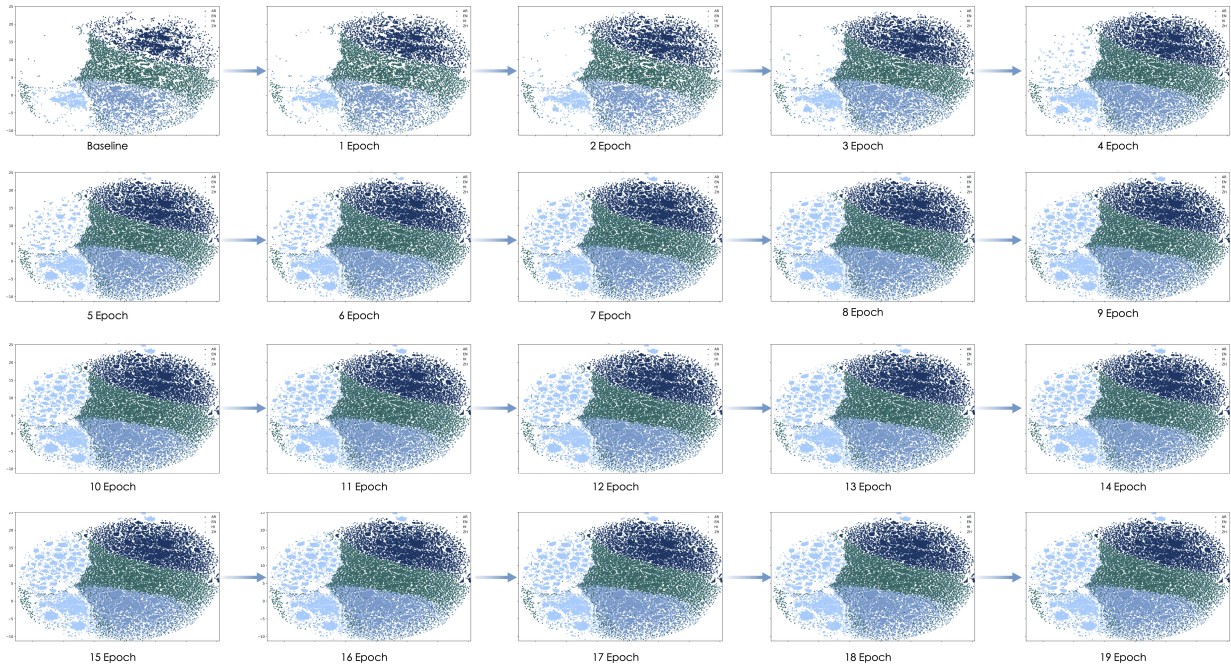

Figure 3: **Temporal evolution of representation geometry under extended continued pretraining.** Joint fit UMAP projections of final layer embeddings from the baseline model and checkpoints from epochs 1 through 19, shown using a shared projection and fixed axis limits. The global organization of representations emerges within the early epochs, with subsequent training primarily redistributing density inside existing regions.

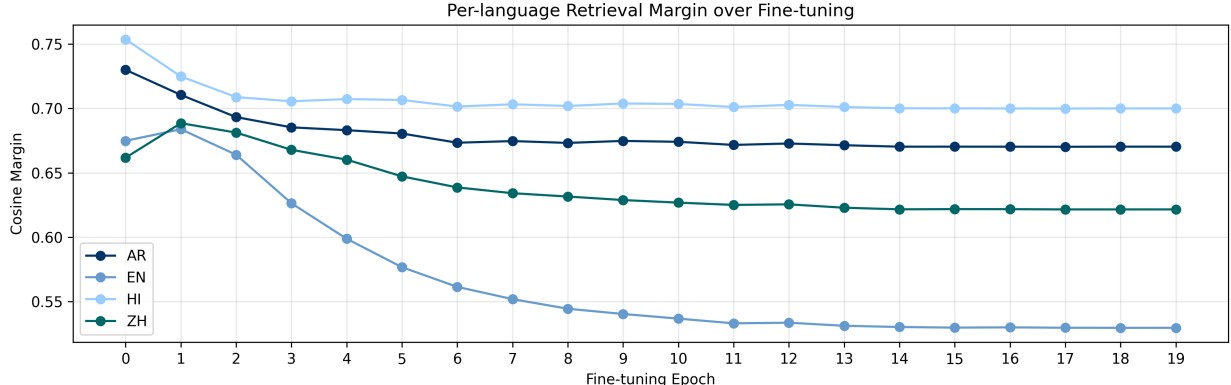

Figure 4: **Per language cosine margin under extended continued pretraining.** Mean centroid based cosine margin for English(EN) , Chinese(ZH), Hindi(HI), and Arabic(AR) from the baseline through epochs 19. Top one language retrieval remains perfect across checkpoints, while the margin decreases and stabilizes after approximately epochs 13.

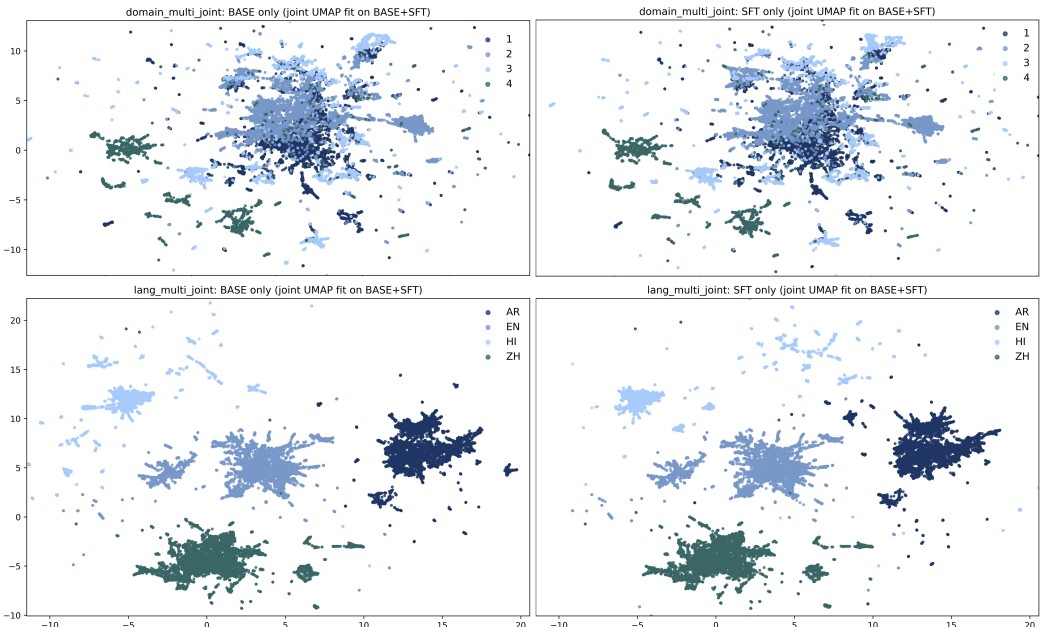

Figure 5: **Geometric effects of full parameter fine tuning across model scale.** Joint fit UMAP projections of final layer embeddings from a 3B model, shown for language conditioned and domain conditioned settings before and after full parameter fine tuning. Language based organization yields distinct regions with density redistribution under training, whereas domain based organization remains largely overlapping with limited geometric change.

### 4.3 Scale Dependence and the Role of Manifold Separability

We apply the same joint fit analysis to a 3B parameter model under identical training conditions to assess whether the observed geometric effects persist with increased capacity. Figure 5 compares language conditioned and domain conditioned settings before and after fine tuning. The qualitative asymmetry observed at 1B remains evident at larger scale. When embeddings are organized by language, the regions remain clearly separated. This organization can already be seen in the pretrained model. Fine tuning mainly adjusts how representations are distributed within each region. In contrast, domain conditioned embeddings continue to occupy a shared and highly overlapping region, even after fine tuning.

Under matched budgets and objectives, adaptation effects vary with the separability of the supported regions in representation space.

**Geometry-defined clusters and cumulative training sets.** For Table 2, we run KMeans with $K = 4$ in the frozen reference space of `BAAI/bge-m3`, then build cumulative training sets by adding clusters in order: C1, C1–2, C1–3, and C1–4. All reported columns in Table 2 are defined in Section 3.4. Here C1–$m$ denotes the cumulative union of clusters $\{1, \ldots, m\}$ in the reference space.

### 4.4 Quantifying Manifold Redistribution under Post Pretraining Adaptation

The qualitative asymmetries observed in Fig. 1 suggest that post pretraining learning does not act uniformly across representation space. To quantify this effect, we compute global and local geometric statistics from last layer embeddings under different training regimes. Average pairwise distances decrease under full parameter updates in both domain-conditioned and language-conditioned settings. The same trend is observed under LoRA, though the magnitude is smaller.

| Cluster | Scale | Training | NLL | global_intra | std | knn_mean | knn_std | knn_cv | knn_q90 |
|---------|-------|----------|-----|--------------|-----|----------|---------|--------|---------|
| C1 | 1B | Full | 2.050 | 42.823 | 6.581 | 27.643 | 6.662 | 0.241 | 36.101 |
| | 3B | Base | 2.085 | 37.851 | 5.419 | 25.738 | 5.988 | 0.233 | 32.692 |
| | 3B | LoRA | 1.923 | 38.111 | 5.593 | 25.356 | 6.155 | 0.243 | 32.354 |
| | 3B | Full | 1.874 | 37.839 | 5.529 | 25.158 | 6.152 | 0.245 | 32.248 |
| C1–2 | 1B | Full | 2.188 | 47.921 | 7.013 | 34.469 | 6.997 | 0.203 | 43.865 |
| | 3B | Base | 2.187 | 42.302 | 5.860 | 31.589 | 6.079 | 0.192 | 39.745 |
| | 3B | LoRA | 2.036 | 42.885 | 5.973 | 31.679 | 6.272 | 0.198 | 39.826 |
| | 3B | Full | 1.997 | 42.429 | 5.946 | 31.294 | 6.275 | 0.201 | 39.594 |
| C1–3 | 1B | Full | 2.139 | 48.796 | 6.342 | 35.157 | 6.441 | 0.183 | 43.469 |
| | 3B | Base | 2.148 | 42.730 | 5.148 | 32.160 | 5.569 | 0.173 | 39.324 |
| | 3B | LoRA | 1.994 | 43.328 | 5.259 | 32.228 | 5.799 | 0.180 | 39.631 |
| | 3B | Full | 1.948 | 42.850 | 5.198 | 31.873 | 5.753 | 0.180 | 39.059 |
| C1–4 | 1B | Full | 2.091 | 48.682 | 6.271 | 35.030 | 6.350 | 0.181 | 42.335 |
| | 3B | Base | 2.091 | 42.512 | 5.010 | 31.970 | 5.414 | 0.169 | 38.639 |
| | 3B | LoRA | 1.940 | 43.177 | 5.171 | 32.066 | 5.650 | 0.176 | 38.840 |
| | 3B | Full | 1.892 | 42.703 | 5.093 | 31.721 | 5.638 | 0.178 | 38.331 |

Table 2: **Representation geometry under progressive cluster aggregation.** Global and local statistics computed from final-layer embeddings. **Training:** Base denotes the pretrained checkpoint; Full denotes full parameter continued pretraining under the standard next token causal language modeling objective; LoRA denotes parameter efficient adaptation (LoRA) under matched optimization settings. For the 1B scale in this table, we report the Full setting only.

Local structure, however, behaves very differently depending on the annotation axis. Under domain conditioned sampling, neighborhoods become progressively more uniform. Both dispersion and tail distances decrease across models and training regimes, as local geometry smooths within an already overlapping space.

Adding languages leads to a looser neighborhood structure, while large distances continue to appear. The same pattern holds across model sizes and adaptation methods, with representations remaining separated by language. Adaptation modifies density within each region rather than pulling them into a single collective environment. As shown in Table 2, progressively aggregating clusters (C1 to C1–4) is associated with smaller relative neighborhood dispersion (`knn_cv`) together with a larger neighborhood scale (`knn_mean`) and a heavier upper tail (`knn_q90`) across scales and adaptation operators.

**Geometric Alignment and the Limits of Specialization**

Language and domain conditioning differ in ways that track their alignment with representation geometry. Adaptation appears most effective when the annotation axis aligns with structure already present in representation space. Language provides such an axis, as it gives rise to coherent regions that fine tuning can refine without altering the global adjacency structure. We emphasize that this paper evaluates representation geometry and objective fit, not downstream task level generalization.

Coarse topical domains overlap extensively and span several latent dimensions, such as style, intent, and register. If labels do not align with separable regions of the manifold, post pretraining learning unfolds within a shared and entangled geometry. Specialization in this regime is therefore weak and largely uniform. Instead, the behavior observed here suggests limits not explained solely by optimization or supervision. They are consistent with a mismatch between supervision labels and the structure already present in representation space.

**Geometric Constraints as a Design Principle**

The importance of geometric constraints is increasingly recognized not only at the data level but also at the architectural level. Recent large scale architecture work has shown that explicitly constraining residual connections to lie on well defined manifolds is essential for stable and scalable training Xie et al. (2026). Although operating at different levels, these architectural results are conceptually aligned with our findings. Both are consistent with the view that large transformations of representation space are difficult to induce under our adaptation setting. Understanding where training data reside within representation space is therefore essential for interpreting both the capabilities and the limits of post pretraining adaptation.

**Implications for Domain Focused and Capacity Constrained Models**

The findings suggest a practical implication for controlling representation consistency in models with limited capacity or a focused use domain. These results provide a geometric perspective that may help interpret reported robustness when training and use remain within geometrically coherent regions, although we do not evaluate deployment reliability directly. Selecting data from dense subregions aligned with a specific slice of the reference space yields more internally consistent representation statistics in our experiments. Regions that are strongly supported by pretraining exhibit more stable directional structure in our measurements.

This notion of maintaining a clean model within a given domain differs from conventional data cleaning or deduplication. The limiting factor is geometric rather than lexical. More consistent representation geometry is observed when adaptation remains within the boundaries of the inherited manifold, rather than relying on surface level similarity. Under this constraint, specialization can emerge more readily in our setting, and compact models exhibit more consistent geometric statistics when operating within regions of representation space that are strongly supported by pretraining.

**Boundary of the Analysis**

The analysis focuses on representations extracted from the final layer and does not trace geometric variation across depth. Prior work indicates that later layers encode task relevant structure, and the present study examines how adaptation redistributes geometry within this representational regime.

The conclusions are specific to adaptation after pretraining under fixed data support, and are stated only in terms of representation geometry and objective level fit rather than downstream task performance. The analysis does not address scenarios in which substantially different data distributions or objectives induce new representational structure. Within the settings considered here, including full parameter continued pretraining updates and LoRA based adaptation, learning appears to be shaped by geometric structure inherited from pretraining. In multi domain deployment under constrained capacity, a single model must span heterogeneous regions of representation space. Under such conditions, specialization correlates with the organization of the inherited representation geometry in the studied setting.

**Model Scale and Scope**

The study considers models at two representative scales, with 1B and 3B parameters. The objective is not to establish state of the art performance, but to isolate geometric constraints that govern adaptation under controlled conditions.

Smaller and mid scale models provide a clearer view of representation geometry, making changes in manifold structure, density redistribution, and separability easier to observe. Across both model scales, and across the two update operators we study (full parameter updates and LoRA), the same qualitative behaviors are consistently observed. This consistency indicates that the reported effects reflect structural properties of representation geometry rather than artifacts of limited capacity.

The central findings may extend beyond the specific scales examined here, particularly in settings that require compact or specialized models due to reliability, efficiency, or deployment considerations.

**Acknowledgments**

The author thanks anonymous reviewers for constructive feedback that improved the clarity of this work.

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

# A  Preliminary Knowledge

This section introduces the representation and geometric quantities used throughout the paper. We describe how representations are extracted from a pretrained decoder only language model, the distance measures used to characterize geometry while separating scale from direction, and the neighborhood statistics that summarize local structure. The definitions provided here are descriptive in nature and establish evaluation objects that are reused in later sections.

## A.1  Sequences, Models, and Layer Representations

Consider a token sequence $x = (x_1, \ldots, x_T)$ of length $T$ and a pretrained decoder only language model $f_\theta$ with parameters $\theta$ and $L$ transformer layers. At each layer $\ell \in \{1, \ldots, L\}$, the model produces a hidden representation $h_\ell(x) \in \mathbb{R}^d$, where $d$ denotes the hidden dimension.

Unless otherwise noted, each sequence $x$ is represented by a single vector from the last layer. In the main experiments we mean pool hidden states over non padding tokens to obtain $z(x) \in \mathbb{R}^d$. We also run a sensitivity check with an alternative readout: the final non padding token.

For a dataset $\mathcal{D} = \{x_i\}_{i=1}^N$, the corresponding set of representations is denoted by

$$\mathcal{Z} := \{z_i = z(x_i)\}_{i=1}^N \subset \mathbb{R}^d. \tag{3}$$

We refer to $\mathcal{Z}$ as an empirical data manifold induced by the pretrained model. This term denotes the observed distribution of samples in representation space, without assuming a parametric form, smoothness, or global manifold regularity beyond what is empirically supported. When comparing training checkpoints, $\mathcal{Z}^{(t)}$ denotes the representation set extracted from checkpoint $t$.

## A.2  Distances, Scale, and Direction

Complementary distance measures are used to separate scale effects from directional structure. For two embeddings $z, z' \in \mathbb{R}^d$, Euclidean distance is

$$d_2(z, z') := \|z - z'\|_2. \tag{4}$$

Euclidean distance depends on both magnitude and direction and therefore captures global scale contraction and norm changes.

To quantify directional relationships, cosine distance is

$$d_{\cos}(z, z') := 1 - \frac{\langle z, z' \rangle}{\|z\|_2 \, \|z'\|_2}, \tag{5}$$

which is invariant to uniform rescaling and isolates differences in direction.

Directional geometry can also be summarized using an $L_2$ metric after normalization. Let $\tilde{z} := z/\|z\|_2$. Then

$$d_2^{\mathrm{dir}}(z, z') := \|\tilde{z} - \tilde{z}'\|_2. \tag{6}$$

This metric provides an equivalent view of angular separation up to a monotone transform, while remaining in Euclidean form. Unless otherwise noted, embeddings are analyzed in raw form so that norm contraction and global scale changes remain observable.

For dataset level summaries, global scale is tracked using the mean norm $\mathbb{E}[\|z\|_2]$ and its dispersion. Global geometry is summarized using expected pairwise distances $\mathbb{E}[d_2]$ and $\mathbb{E}[d_{\cos}]$. Because exact pairwise computation scales as $O(N^2)$, these expectations are estimated by random subsampling of $M$ index pairs under a fixed random seed shared across checkpoints and settings. We sample pairs with $i \neq j$ to exclude self pairs.

### A.3   k Nearest Neighbor Neighborhoods and Local Density Proxies

Local structure is described using $k$ nearest neighbor neighborhoods under Euclidean distance in $\mathbb{R}^d$. For each representation $z_i \in \mathcal{Z}$, the neighborhood $\mathcal{N}_k(z_i) \subset \mathcal{Z} \setminus \{z_i\}$ contains its $k$ closest neighbors.

The mean neighbor distance is defined as

$$\rho_k(z_i) := \frac{1}{k} \sum_{z_j \in \mathcal{N}_k(z_i)} \|z_i - z_j\|_2. \tag{7}$$

This quantity serves as a local density proxy. Smaller values correspond to denser regions of $\mathcal{Z}$, whereas larger values indicate sparser or more boundary like locations. The statistic $\rho_k$ is not intended as a parametric density estimator. Instead, it provides a robust geometric summary of neighborhood tightness in high dimensional spaces without introducing kernel bandwidths or smoothness assumptions.

Neighborhood structure at the dataset level is summarized by distributional statistics over $\{\rho_k(z_i)\}_{i=1}^N$. These include the mean and standard deviation, the coefficient of variation $\mathrm{CV} := \sigma/\mu$, and upper tail quantiles such as the 90th percentile. Together, these summaries distinguish uniform redistribution within a shared region from persistent long range neighborhoods that arise when representations occupy separated submanifolds.

### A.4   Quantile Partitions for Sampling in Reference Space

We restate the sampling partitions for completeness. All center and periphery subsets are defined in a frozen reference space $\mathcal{Z}_{\mathrm{ref}} = \{u_i\}_{i=1}^{N_{\mathrm{pool}}}$. Define

$$\rho_k^{\mathrm{ref}}(u_i) := \frac{1}{k} \sum_{u_j \in \mathcal{N}_k^{\mathrm{ref}}(u_i)} \|u_i - u_j\|_2,$$

where $\mathcal{N}_k^{\mathrm{ref}}(u_i) \subset \mathcal{Z}_{\mathrm{ref}} \setminus \{u_i\}$ contains the $k$ nearest neighbors of $u_i$ under Euclidean distance in the reference space. Let $Q_\alpha^{\mathrm{ref}}$ denote the $\alpha$-quantile of $\{\rho_k^{\mathrm{ref}}(u_i)\}_{i=1}^{N_{\mathrm{pool}}}$. Then

$$\mathcal{D}_{\mathrm{center}} := \{x_i : \rho_k^{\mathrm{ref}}(u_i) \leq Q_{\alpha_c}^{\mathrm{ref}}\}, \qquad \mathcal{D}_{\mathrm{periph}} := \{x_i : \rho_k^{\mathrm{ref}}(u_i) \geq Q_{\alpha_p}^{\mathrm{ref}}\}.$$

All reported geometry is computed in the decoder only model space and is never used for sampling.

## B   Consistency Check Between Reference and Pretrained Spaces

Dataset partitions are defined in a frozen reference embedding space. As a simple consistency check, we report how neighborhood structure in the reference space relates to the pretrained model representation space before any adaptation.

We evaluate 10,000 sequences and run two checks.

**Agreement in density ordering.**   For each sequence, we compute a mean $k$ nearest neighbor distance in (i) the reference space, $\rho_k^{\mathrm{ref}}(u_i)$, from `BAAI/bge-m3`, and in (ii) the pretrained model space, $\rho_k(z_i)$, from the final-layer representations of `meta-llama/Llama-3.2-1B`. We use Euclidean $k$NN distances for density ordering in both spaces. We then compare the induced rankings using Spearman correlation. With $k = 10$, we obtain

$$\rho_S = 0.714, \quad p < 10^{-10}.$$

Here $\rho_S$ denotes Spearman's rank correlation coefficient and $p$ denotes the associated two-sided p value. This indicates that samples ranked as more central or more peripheral in the reference space tend to be ranked similarly in the pretrained model space.

**Agreement in pairwise proximity.** Computing a full $N \times N$ distance matrix is $O(N^2)$ and is prohibitive at scale. We therefore approximate distance-matrix agreement by Monte Carlo subsampling. We sample $M_{\text{agree}} = 200,000$ random pairs (seed 42) and compute cosine distances in both spaces. We use cosine distance to match the main paper's directional metrics. We observe moderate positive correlation (Spearman $\rho_{\text{S}} = 0.385$, Pearson $r = 0.395$, $p < 10^{-10}$), suggesting partial preservation of relative proximity.

**Summary.** Overall, the reference space aligns with the pretrained model space in density ranking and shows moderate consistency in pairwise proximity. These results support using the reference encoder as an instrument for defining a fixed sampling rule, while all geometric measurements and conclusions in the paper are computed in the pretrained and adapted model space.

## C  Additional Results on Language Retrieval Margin

For a sample embedding $\tilde{z}$ (normalized), let $\mathcal{I}_y$ be the index set of evaluation samples whose language label is $y$. Define the (unnormalized) centroid as $\bar{c}_y := \frac{1}{|\mathcal{I}_y|} \sum_{i \in \mathcal{I}_y} \tilde{z}_i$ and the normalized centroid as $c_y := \bar{c}_y / \|\bar{c}_y\|_2$. We define the cosine margin as $m(\tilde{z}) := \cos(\tilde{z}, c_y) - \max_{y' \neq y} \cos(\tilde{z}, c_{y'})$. Table 3 summarizes cosine margin statistics for language identification, computed using centroid based representations, across the baseline model and checkpoints epoch 1 through epoch 19. Reported values include the mean and standard deviation of the margin over the full test set, as well as results for individual language subsets (EN, ZH, HI, AR).

| Checkpoint | Overall | English | Chinese | Hindi | Arabic |
|---|---|---|---|---|---|
| Baseline | 0.705±0.077 | 0.675±0.046 | 0.662±0.104 | 0.754±0.058 | 0.730±0.043 |
| Epoch 1 | 0.702±0.074 | 0.684±0.059 | 0.689±0.102 | 0.725±0.071 | 0.710±0.041 |
| Epoch 2 | 0.687±0.075 | 0.664±0.070 | 0.681±0.098 | 0.709±0.071 | 0.693±0.041 |
| Epoch 3 | 0.671±0.082 | 0.626±0.087 | 0.668±0.094 | 0.706±0.074 | 0.685±0.040 |
| Epoch 4 | 0.662±0.087 | 0.599±0.090 | 0.660±0.093 | 0.707±0.074 | 0.683±0.039 |
| Epoch 5 | 0.653±0.092 | 0.577±0.089 | 0.647±0.093 | 0.707±0.076 | 0.681±0.039 |
| Epoch 6 | 0.644±0.093 | 0.561±0.088 | 0.639±0.095 | 0.701±0.075 | 0.673±0.037 |
| Epoch 7 | 0.641±0.095 | 0.552±0.085 | 0.634±0.097 | 0.703±0.075 | 0.675±0.037 |
| Epoch 8 | 0.638±0.097 | 0.544±0.083 | 0.632±0.098 | 0.702±0.075 | 0.673±0.036 |
| Epoch 9 | 0.637±0.097 | 0.540±0.080 | 0.629±0.098 | 0.704±0.074 | 0.675±0.035 |
| Epoch 10 | 0.635±0.098 | 0.537±0.079 | 0.627±0.098 | 0.703±0.073 | 0.674±0.035 |
| Epoch 11 | 0.633±0.098 | 0.533±0.078 | 0.625±0.098 | 0.701±0.072 | 0.672±0.034 |
| Epoch 12 | 0.634±0.098 | 0.534±0.077 | 0.625±0.098 | 0.703±0.072 | 0.673±0.034 |
| Epoch 13 | 0.632±0.098 | 0.531±0.076 | 0.623±0.099 | 0.701±0.072 | 0.671±0.034 |
| Epoch 14 | 0.631±0.098 | 0.530±0.076 | 0.622±0.099 | 0.700±0.072 | 0.670±0.034 |
| Epoch 15 | 0.630±0.098 | 0.530±0.076 | 0.622±0.100 | 0.700±0.071 | 0.670±0.034 |
| Epoch 16 | 0.630±0.098 | 0.530±0.076 | 0.622±0.100 | 0.700±0.071 | 0.670±0.034 |
| Epoch 17 | 0.630±0.098 | 0.530±0.075 | 0.622±0.100 | 0.700±0.071 | 0.670±0.034 |
| Epoch 18 | 0.630±0.098 | 0.530±0.075 | 0.622±0.100 | 0.700±0.071 | 0.670±0.034 |
| Epoch 19 | 0.630±0.098 | 0.530±0.075 | 0.622±0.100 | 0.700±0.071 | 0.670±0.034 |

Table 3: **Centroid-based cosine margin for language retrieval across checkpoints.** Mean ± standard deviation of the cosine margin computed on a fixed held-out evaluation set ($N_{\text{test}} = 2000$, 500 samples per language). Four language centroids are computed once from the baseline model and kept fixed across all checkpoints. Statistics are reported for the full evaluation set and for each language subset (EN, ZH, HI, AR).

