# OpenReview forum: "What Constrains Adaptation After Pretraining? Representation Dynamics Under Inherited Data Manifolds"
_TMLR — Rejected by TMLR_

### Review · Reviewer_Bi6a · 2026-01-29

**Summary Of Contributions:**

The paper studies representations of pretrained large language models and analyzes how their geometric properties evolve under fine-tuning and distillation. The stated goal is to assess whether limitations in fine-tuning performance arise from geometric constraints inherited from pretraining, rather than from deficiencies in optimization or supervision. To this end, the paper analyzes representations from the final layer of the language model and introduces an external sentence embedding space in which dataset density is measured via neighborhood statistics.

**Strengths**: In contrast to much prior work that focuses on improving performance through architectural or optimization changes, this paper highlights the role of data on finetuning performance. It aims to characterize how the structure of learned representations constrains adaptation, which is a valuable and underexplored angle.

**Weaknesses**:

1.	The paper claims experimental results that are not supported by experiments presented in the main text.
2.	The experimental setup is insufficiently specified, making the results difficult or impossible to reproduce.
3.	Many claims are phrased imprecisely, with key terms and quantities left undefined.
4.	The paper lacks a clearly articulated hypothesis or theoretical framework that would explain the reported observations.

***************
**On 1**:  Missing or unsupported experimental evidence

The abstract claims that “data location in representation space strongly constrains what can be learned” and that “models trained on data drawn from peripheral or highly overlapping regions tend to generalize poorly, even when the training setup is otherwise unchanged.”

However, I was unable to locate an experiment in the paper that directly supports these statements. Although the paper defines center and periphery subsets via neighborhood density, no experiment is presented in which models are trained on center versus peripheral data and evaluated on held-out or complementary regions to measure generalization. Neither Table 1 nor Table 2 nor Figures 1–5 report such a comparison. Instead, these results primarily report representation geometry statistics (e.g., k-nearest-neighbor distances) rather than generalization performance.
Similarly, the paper lists several baseline sampling strategies (e.g., random sampling, quality filtering, curriculum variants), but no experiment compares geometry-conditioned sampling against these baselines, nor are results for them reported.

**On 2** Insufficient specification and lack of reproducibility

Key components of the experimental setup are underspecified or missing entirely. Examples include:

- The implementation section contains unresolved placeholders (e.g., “[R]”, “[model name]”), and although the paper claims to report mean ± standard deviation, no such statistics appear in the results of the main text.

- The identities of the “1B” and “3B” pretrained models are not given.

- The “large public text distribution” from which data are drawn is not specified.

- The supervised fine-tuning task (e.g., “language-conditioned fine-tuning”) is not clearly defined.

- Data budgets, input formatting, and optimization schedules are referenced but not described. (““All SFT runs use the same data budget, input formatting, and optimization schedule across geometric conditions.” )

- The sentence embedding model (“BGE”) used to define dataset geometry is not specified.

- The choice of quantile thresholds for defining center and periphery subsets is not given.

- Hyperparameters for SFT, LoRA, and distillation are said to follow “standard practice” without further detail.

- The choice of k for k-nearest neighbor statistics is not given

- In Table 2, what are the domains x1—x4?

Taken together, these omissions prevent reproduction of the experiments and weaken the empirical claims.

**On 3** Imprecise and vague claims

Several statements in the paper lack clearly defined notions or measurements. For example, the introduction states:
“We observe the same pattern across model sizes and across both fine tuning and distillation. Training that emphasizes boundary regions leads to lasting differences in generalization.” -- It is unclear what “the same pattern” refers to, how “differences in generalization” are quantified, or which specific geometric properties of the manifold are being measured. Similar imprecision appears throughout the paper, making it difficult to assess the strength and scope of the conclusions.

**Audience:**

No

**Audience Explanation:**

The motivation to study how data selection and representation geometry influence post-pretraining adaptation is well aligned with current interests in the community. However, the results presented in this paper remain preliminary and largely descriptive. Key claims are not clearly supported by identifiable experiments, and the lack of precise definitions and reproducible setups limits the extent to which the findings can be validated or built upon. As a result, it is difficult to assess the contribution or derive actionable insights, which reduces the paper’s impact and relevance for the community in its current form.

**Claims And Evidence:**

No

**Claims Explanation:**

Several statements in the paper are insufficiently substantiated by clearly identifiable evidence. In particular, the abstract and discussion refer to experimental findings—such as degraded generalization when training on peripheral or highly overlapping regions—that I was unable to locate unambiguously in the manuscript.

More generally, imprecise notation, missing definitions, and underspecified experimental choices make it difficult to trace conclusions back to concrete experimental results. As a result, it is challenging to evaluate the correctness of the claims or to reproduce the reported findings.

**Requested Changes:**

I suggest that the authors revise the manuscript to improve clarity and conciseness, and to explicitly identify and describe the experiments that support the claims made in the abstract and throughout the paper. In particular, any experiments underlying statements about center versus peripheral data, generalization behavior, or specialization should be clearly presented and linked to specific results.

In addition, all implementation details necessary for reproduction should be provided, including model identities, datasets, training setups, hyperparameters, and the precise definitions of all notation used in the tables and figures.

Finally, the paper would benefit from a clearer statement of the hypothesis being tested, ideally accompanied by a theoretical motivation or framework. At present, the work primarily reports experimental observations and post-hoc interpretations; articulating an explicit hypothesis would help clarify the intended contribution and strengthen the empirical evaluation.

---

### Review · Reviewer_5coQ · 2026-02-01

**Summary Of Contributions:**

The paper analyses how the representations geometry change by supervised finetuning (SFT) stage for language modeling (LM) to understand the failures in LMs generalization (if the source is geometric constraints from pretraining stage or SFT/optimization themselves).
Authors propose to perform controlled analysis where data for supervised finetuning are sampled from different geometric regions (dense or peripheral) or following different policy and check how this affects the representations geometry after SFT. The conclusion authors get is "Models trained on data drawn from peripheral or highly overlapping regions tend to generalize poorly".

Strengths:
- motivation and idea to look into the data and geometric structure + analyse how they change during post training stages (SFT)
- proposal on evaluation setup to fix everything except how data sampled and consider only train data as the factor

Weaknesses:
- writing and explanation of results
- empirical setup is flawed
- results cannot be reproduced
- claims are not supported
- missing experiments / comparisons announced in the paper

**Audience:**

No

**Audience Explanation:**

While the motivation is strong for TMLR and the audience, the current state of the paper doesn't provide strong supported empirically (or theoretically) claims which will be helpful. The opinion here is mainly driven by the current version which I found to be incomplete. After substantial revision and more empirical detailed supporting results it could be interesting to the community for their findings.

**Broader Impact Concerns:**

No concerns as the paper analyses geometric structure of the models trying to improve domain adaptation and robustness of the models.

**Claims And Evidence:**

No

**Claims Explanation:**

Please find below extended explanation on the weaknesses section and why I believe in the current form the claims are not supported:
- I believe the paper needs substantial rewriting as many places has duplicated thoughts and conclusions and thus in many places I see repetition of the same message
- While authors state that they consider SFT, in fact it is continued pretraining (as no any labels, instructions are used)
- All hyperparameters, model details (arch, etc.), optimizer, data source used in the analysis, even critical quantiles are not specified. Overall from the current state of the description in the paper people cannot reproduce any results and analysis.
- Claims are not supported:
  - baselines specified in page 5 are not provided throughout the paper
  - generalization / specialization is not provided either (maybe I missed it?)
  - empirical discussion of results for language and domain are very unclear to me and how they related exactly to the main claims
  - with missing details on how exactly experiment is done it is hard to conclude that claim is supported
- Overall, the beginning of the paper establishes good motivation and reasoning, but as it goes to validation and analysis I am completely lost in what is going on, what exactly setting is done and how it is related to the main claim from the abstract. A lot of experiment results initially announced are not provided. Appendix also doesn't give any extra details to resolve this.

**Requested Changes:**

Please find below detailed comments which should also cover the issues with claims I see in the current form of the paper.
- (formatting) literature style should be in brackets when it is a reference, and w/o if you use it as part of a sentence. Usage of E19, E13 is not introduced (though I get it).
- "Adaptation after pretraining modifies model behavior while operating on representations that are already established." I am not sure about this statement. It still may change representations a lot as we finetune the whole model often in post-training. I guess you need to clarify here what adaptation means exactly.
- "From a geometric perspective, adaptation after pretraining redistributes density within an inherited representation space, while the large scale structure remains unchanged" - first of all I can see this as a very old reference, which may not be true for large scale models. But then if this is universally true then what is the authors' contribution if this prior work already showed the structure authors try to show? Similar issue with referencing "Across these approaches, adaptation does not introduce fundamentally new representational structure" - recent work and the it questions authors contribution. My suggestion: contrast better what is the difference with your work and what you bring compared to prior works (even if it is confirming the prior results for modern models). Overall the whole section on related works should place your work compared to prior to clearly show what you bring to the table.
- SFT setup is using supervised data. But I see that you are still training for next token prediction only w/o any labels (it is not described in the paper), so there are no supervised data at all. This contradicts the initial setting and the whole main claims in abstract.
- "sequence x is represented by the hidden state at the final non padding token" - I really don't get why you take the last token in sequence representation. This doesn't make sense to me at all. The last representation may contain whole sequence info, but it should be dominated by the current token as it is the task of next token prediction (not supervised training per info I see in the paper right now). I can only guess that you may consider some supervised finetuning with simple answers (yes, no) and that is why the last token will be the answer, thus representation for the last token can make sense.
- "the emphasis is not on maximizing absolute task performance." - while I agree with the overall principle here, accuracy must be reported, as otherwise you may end up analysing the model with very bad quality and thus representations may not reflect real models at all. So accuracy of SFT must be reported for both in domain data and out of domain.
- All exact details on data, models, hyperparameters, optimizers, sampling data sizes, quantiles, anything else for reproducibility must be provided.
- Baselines listed on page 5 are not discussed, shown and compared with.
- Results for 1B only are only shown for language axes but no domains. So the statement "The qualitative asymmetry observed at 1B remains evident at larger scale." is not true as results are missing. Moreover I don't get what you mean and do exactly for different axes (language or domain) in terms of model finetuning. Are language and domain prediction your supervised tasks? If not then it doesn't make sense at all to me why to look into their geometric structure: if I finetune, say for Q&A, I would expect that the representations between languages will remain clustered well, while for domains not. I did not get the story and explanations for language and domain axes in the paper.
- "Both panels rely on the same UMAP projection" is it done jointly in the end? Can you be more specific how exactly you do it as later in text it is used "joint fit UMAP".
- "These observations indicate that the effectiveness of post pretraining adaptation depends critically on the existing separability of the data manifold." Can you explain this and how you end up with this conclusion? I don't get the reasoning behind (likely due to missing many details on exact setup and connection between language/domain axis and initial description of peripheral and center data).
- Table 2 - what is x1, x2, x3, x4?
- Results 1B, 3B and Discussion sections are repeating the main messages, I would recommend rewriting and maybe stating particular property you found first and then show results confirming it e.g.
- TBH many pieces of discussion section for me is speculation w/o supporting claims and it is for some reason hard to read to get the main message you are trying to say.

It may be that I didn't fully understand the paper and main claims are supported, but at least more details should be provided. Also I still suggest polishing the key contributions and results with clear messages so that other people also do not have similar confusion as I am right now.

---

### Review · Reviewer_FsoW · 2026-02-04

**Summary Of Contributions:**

This paper investigates the geometric constraints imposed by pretraining on subsequent model adaptation (SFT, LoRA, and distillation). The authors argue that adaptation acts as a "density redistribution" within an inherited manifold rather than a fundamental restructuring of the representation space. By utilizing local density proxies ($\rho_k$) and k-nearest neighbor statistics, they demonstrate that data location (center vs. periphery) and the alignment between supervision labels and existing geometric clusters (language vs. domain) dictate the success of adaptation.

**Key Strengths:**

- Novel Perspective: Shifting the focus from optimization/scaling to the geometric organization of the inherited data manifold.


- Rigid Experimental Control: Controlling for architecture, capacity, and supervision to isolate "data geometry" as the sole independent variable.


- Scalability Analysis: Validating that these geometric constraints persist across model scales (1B to 3B parameters).

**Key Weaknesses:**


- Logical Inconsistency in Terminology: The authors frequently claim that representation structure remains "unchanged" , while simultaneously reporting data that shows the structure becoming significantly more distinct or "sharpened".


- Limited Scope: The analysis is restricted to the final layer, ignoring how these manifolds evolve through the depth of the transformer.

**Audience:**

Yes

**Audience Explanation:**

The findings regarding center-periphery sampling and the limits of domain-specific specialization are highly relevant to researchers working on model efficiency, data selection, and foundation model adaptation. Understanding that pretraining "locks in" certain geometric properties can help practitioners better anticipate where fine-tuning will succeed or fail.

**Broader Impact Concerns:**

None. This is a fundamental study of representation geometry and does not raise immediate ethical concerns.

**Claims And Evidence:**

Yes

**Claims Explanation:**

The empirical evidence provided in Tables, along with the visualizations, is comprehensive and supports the general trend of density redistribution. However, the interpretation of this evidence is where the paper need to be improved. For instance, the authors state that "overall geometric structure remains largely unchanged" , yet Table 1 explicitly shows $d_{cos}$ increasing and $d_2^{dir}$ rising. This is a measurable change in directional structure. While the topological order of clusters may persist, the geometry is clearly being sharpened. The claims are supported by the data, but the narrative needs to be more precise about what exactly is "unchanged" versus what is "refined."

**Requested Changes:**

- Clarify "Unchanged" vs. "Sharpened" Structure: The authors must reconcile the claim that "structure remains unchanged" with the empirical finding that angular separation increases. It is more accurate to describe this as "structural persistence with directional sharpening" rather than an "unchanged" manifold.

- Define "Restructuring": To support the claim that adaptation does not restructure space, the authors should provide a conceptual or hypothetical example of what "restructuring" would look like in their metric space.

---

### Review · Reviewer_D5SR · 2026-02-05

**Summary Of Contributions:**

This paper investigates the geometric constraints that a pretrained model’s representation space imposes on subsequent adaptation (fine-tuning). By partitioning data into "central" (high-density) and "peripheral" (low-density) regions of the inherited manifold, the authors demonstrate that generalization is not merely a function of data volume or optimization, but of geometric alignment. They find that adaptation primarily redistributes density within fixed manifolds rather than reorganizing them. Consequently, specialization is highly effective for well-separated clusters (like different languages) but fails for overlapping, "entangled" regions (like topical domains).

Strengths:

S1: Novel Geometric Perspective: The paper shifts the focus from traditional scaling laws to the "geometric inheritance" of models. It provides a compelling argument that the success of fine-tuning is predestined by how well the new task labels align with the pre-existing structure of the pretrained embedding space.

S2: Principled Experimental Design: The use of a "reference space" (BGE embeddings) to compute density proxies and create "Center vs. Peripheral" data splits is a rigorous way to isolate geometry as a variable, independent of model size or training duration. (Though see W4)

S3: Scale-Invariant Findings: By demonstrating that these geometric constraints persist across both 1B and 3B parameter models, the research finds that simply increasing model capacity does not "solve" the issues of overlapping manifolds or poor peripheral generalization. (Though see W5)

S4: Strong Theoretical Grounding: The work demonstrates a thorough reliance on and synthesis of previous literature, effectively bridging classical manifold learning theory with modern transformer adaptation research to provide a robust conceptual framework.

Weaknesses

W1: Unfinished "Draft" Quality: The manuscript is riddled with raw LaTeX-style placeholders (e.g., [model name], [R] repetitions, and [link]). This lack of specificity regarding the actual base models used makes it difficult for other researchers to replicate or fully trust the empirical results.

W2: Poor Mathematical and Narrative Flow: Notation elements are frequently used before they are formally defined (e.g., density metrics and quantile variables appearing in abstracts and figures before the methodology section). This creates a disjointed reading experience that obscures the technical logic. In particular, the main findings and claims are presented very late in the paper, and with insufficient clarity.

W3: Inconsistent Stylistic Execution: The citation style is erratic, with numerous "orphaned" parentheticals and un-truncated bibliographies (like the 100+ author Bommasani citation). This, combined with the "weird" formatting of author names within the text, detracts from the professional quality of the findings.

W4: Proxy Bias (The "BGE" Problem): The study assumes BGE embeddings are a neutral "ground truth" for data geometry. However, using one pretrained model to measure another risks circular logic; the observed "constraints" may simply be the specific biases and density distributions of the BGE model itself. In particular, findings regarding language vs other domains may be heavily impacted.

W5: Limited Scope: The evaluation is restricted to small-scale models (1B and 3B) and a single embedding reference. It remains unproven whether these geometric constraints persist in larger models (70B+) or if the definitions of "Center" and "Periphery" would shift significantly using different architectural references.

**Audience:**

Yes

**Audience Explanation:**

The core hypothesis is of high interest to the TMLR community. The paper's approach provides a theoretical framework for understanding why fine-tuning often yields unreliable generalization. If the authors can address the 'BGE bias' and provide a more rigorous, model-internal analysis, the findings would contribute significantly to our understanding of the relationship between pretraining manifolds and task-specific adaptation. Moreover, there may be implications for topics of great contemporary interest such as post-training of LLMs, its safety and alignment.

**Claims And Evidence:**

No

**Claims Explanation:**

While the paper proposes a compelling and potentially high-impact theory regarding geometric inheritance, the current evidence is unconvincing due to the lack of model specificity (placeholders), unclear due to the premature use of mathematical notation and stylistic choices, and potentially inaccurate because it fails to account for the inherent biases of the BGE embedding proxy used to define the manifold.

**Requested Changes:**

Critical for acceptance:

C1: Address the BGE Proxy Bias: The authors must provide evidence that the "Central vs. Peripheral" findings are not an artifact of the BGE model’s specific biases.

C2: Removal of All Placeholders: The manuscript must be finalized. All raw LaTeX/draft placeholders must be replaced with concrete data, specifically identifying the base model architecture (replacing [model name]) and the number of experimental repetitions (replacing [R]).

C3: Formalize Notation and Sequence: The mathematical narrative must be repaired so that every variable is formally defined at or before its first use. This is particularly important for the density metrics, which are currently referenced in the abstract and early figures without definition.

C4: Stylistic and Bibliographic Overhaul: Standardize the citation format to ensure names are grammatically integrated into the text. Truncate the massive "Bommasani et al." bibliography entry and other over-length references to improve readability.

C5: Clarity of Key Claims and Contributions: The authors should revise the early sections to explicitly state their main findings as the primary theoretical takeaways, rather than letting them emerge as secondary observations in the results section.

Recommended:

R1: Discussion of Safety and Alignment: While a full experimental branch may be out of scope, the authors are encouraged to add a discussion section on how "geometric entanglement" and "peripheral unreliability" might impact the robustness of post-training safety guardrails and alignment (RLHF/DPO). This may greatly increase the interest in, and impact of, their work.

R2: Scale Expansion: Adding evaluation data for a larger model (e.g., 7B or 8B) would significantly strengthen the claim that these geometric constraints are a fundamental property of transformer manifolds and not a limitation of "small" (1B-3B) models.

R3: Intrinsic Dimensionality Analysis: To move beyond the limitations of 2D visualizations (t-SNE/UMAP), the authors could strengthen the "overlap" claim by calculating the intrinsic dimensionality or manifold capacity of the entangled regions.

R4: Sensitivity of $k$: Provide a brief analysis of how the choice of $k$ in the k-NN density formula (Equation 1) affects the results, ensuring the "Center/Periphery" split is robust to parameter tuning.

---

### Decision · Action_Editor_HVFi · 2026-03-03

**Recommendation:** Reject

**Audience:**

No

**Audience Explanation:**

While the paper's topic is relevant for TMLR, the presentation and state of the paper doesn't provide empirically supported claims that may be of interest to the TMLR audience.

**Claims And Evidence:**

No

**Claims Explanation:**

The reviewers found that the claims in the paper are not clearly nor precisely formulated to support the experimental evidence in the paper.  Despite the authors comments and changes, the reviewers maintained that there are terms and concepts that are not clearly defined.  Overall, the claims are not empirically nor theoretically supported.